# Dose-response analysis between hemoglobin A1c and risk of atrial fibrillation in patients with and without known diabetes

Huilei Zhao[1‡], Menglu Liu[2‡], Zhifeng Chen[3], Kaibo Mei[4], Peng Yu[5], Lixia Xie[6]*

1 Department of Anesthesiology, The Third Hospital of Nanchang, Nanchang, Jiangxi, China, 2 Department of Cardiology, The Seventh People's Hospital, Zhengzhou, Henan, China, 3 Fuzhou University of International Studies and Trade, Fuzhou, Fujian, China, 4 Department of Anesthesiology, Shangrao People's Hospital, Shangrao, Jiangxi, China, 5 Department of Endocrinology, The Second Affiliated Hospital of Nanchang University, Nanchang, Jiangxi, China, 6 Department of Respiratory and Critical Care Medicine, The Second Affiliated Hospital of Nanchang University, Nanchang, Jiangxi, China

‡ HZ and ML are co-first authors.
* xielixia2030@163.com

**Data Availability Statement:** All relevant data are within the manuscript and its Supporting Information files.

## Abstract

### Background

The relationship between serum hemoglobin A1c (HbA1c) and atrial fibrillation (AF) or post-operative AF (POAF) in coronary artery bypass (CABG) patients is still under debate. It is also unclear whether there is a dose-response relationship between circulating HbA1c and the risk of AF or POAF.

### Methods and results

The Cochrane Library, PubMed, and EMBASE databases were searched. A robust-error meta-regression method was used to summarize the shape of the dose-response relationship. The RR and 95%CI were using a random-effects model. In total, 14 studies were included, totaling 17,914 AF cases among 352,325 participants. The summary RR per 1% increase in HbA1c was 1.16 (95% CI: 1.07–1.27). In the subgroup analysis, the summary RR was 1.13 (95% CI: 1.08–1.19) or 1.12 (95% CI: 1.05–1.20) for patients with diabetes or without known diabetes, respectively. The nonlinear analysis showed a nonlinear (P_nonlinear = 0.04) relationship between HbA1c and AF, with a significantly increased risk of AF if HbA1c was over 6.3%. However, HbA1c (per 1% increase) was not associated with POAF in patients with diabetes (RR: 1.13, P = 0.34) or without known diabetes (RR: 0.91, P = 0.37) among patients undergoing CABG.

### Conclusion

Our results suggest that higher HbA1c was associated with an increased risk of AF, both in diabetes and in without diabetes or with unknown diabetes. However, no association was found between HbA1c and POAF in patients undergoing CABG. Further prospective studies with larger population sizes are needed to explore the association between serum HbA1c level and the risk of POAF.

**Competing interests:** The authors have declared that no competing interests exist.

## Introduction

Atrial fibrillation (AF) is the most common cardiac arrhythmia found in clinical practice and is associated with an increased risk of morbidity [1]. As we known, there is strong independent association between diabetes and AF. Moreover, studies also showed fasting glucose levels were positively and independently correlated with incident AF in patients with diabetes, which suggested the important role of glycemic control on preventing the AF. However, results from Action to Control Cardiovascular Risk in Diabetes trial showed that intensive glycaemic control in people with diabetes have no significant effect on the rate of new-onset atrial fibrillation and cardiovascular outcomes.

Hemoglobin A1c (HbA1c), a major component of hemoglobin-glucose adducts, reflects the average blood glucose level within the prior 2 to 3 months and has been a well-accepted biomarker for glycemic management in diabetes patients over the past several decades[2]. In recent years, an elevated level of HbA1c has been shown to be linked to an increased risk of cardiovascular diseases (e.g., myocardial infarction and AF)) in patients with and without diabetes [3]. The association between HbA1c and AF has been investigated in previous studies. However, the evidence is inconsistent [4–9]. In the Atherosclerosis Risk in Communities cohort, Huxley et al[5] found that high HbA1c was an independent risk factor for incident AF in patients with and without diabetes. A meta-analysis also found that serum levels of HbA1c in diabetes patients with AF were higher than those in diabetes patients without AF (standardized mean difference = 0.67)[10]. Conversely, the results of the Women's Health Study showed that there was no association between any category of HbA1c level (4.84%-5.0%, 5.0%-5.19%, >5.19%) and risk of AF after adjusting for potential covariates[11]. Recent, several new studies with large population were published, still with inconsistent results[12, 13]. Moreover, several new research articles have reported that a higher serum HbA1c could reduce new-onset AF after coronary artery bypass grafting (CABG) surgery [14, 15]. Given this background, we conducted a dose-response meta-analysis to i) quantitatively investigate the dose-specific relationship between HbA1c and AF. ii) estimate whether serum HbA1c could reduce post-operation AF (POAF) after CABG.

## Methods

### Literature search

This work has been performed according to PRISMA guidelines (http://www.prisma-statement.org; **S1 Table**). We systematically searched the Cochrane Library, PubMed, and Embase databases for eligible studies before March 10, 2019.Two groups of keywords (linked to hemoglobin A1c, and AF, respectively) were combined using the Boolean operator "and". **S2 Table** provides a detailed description of the search strategy in the aforementioned electronic databases. In addition, we searched the reference lists of two review or other relevant publications to identify further studies. No language restrictions were applied in the whole literature search.

### Study selection

Studies were considered eligible if they: (1) designed as randomized controlled trials (RCTs) or observational cohorts; (2) reported the impact of hemoglobin A1c and AF; (3) made available a quantitative measure of hemoglobin A1c and the number of AF cases in each hemoglobin A1c category for the dose-response analysis. For multiple publications/reports created from the same data, the studies with the longest follow-up period or the largest number of AF cases were included. In addition, certain publication types (e.g., reviews, editorials, letters, conference abstracts, and animal studies), or studies with insufficient data were excluded from this analysis. Post-hoc analyses of RCTs can be equivalent to observational studies.

## Data extraction and quality assessment

For each study, the basic characteristics were extracted, mainly including the first author, publication year, geographical location, study type, participants (sex, age, and sample size), duration of follow-up, adjustments for confounders, AF type (on-set or recurrence) methods of measuring hemoglobin A1c levels, categories of hemoglobin A1c and adjusted risk ratios (RRs) (categorical or continuous) with its 95% confidence intervals (CIs). If both unadjusted and adjusted RRs existed in one study, we extracted the most completely adjusted one. For multiple publications/reports with the same data, we included the article with the most recent or the largest number of participants.

We used the Newcastle-Ottawa quality assessment scale (NOS) to evaluate the quality for all included studies[4]. Post hoc analysis of RCTs was treated as cohort studies to achieve quality assessment[16]. The validated NOS items with a total of 9 stars involved three aspects including the selection of population, the comparability of study, and the assessment of the outcome. In this meta-analysis, a NOS score of ≥6 stars was regarded as high-quality, otherwise, as low-quality studies[5].

## Statistical analyses

Both linear and nonlinear models were performed. We calculated study-specific RR (hemoglobin A1c per 1% increase) and 95% CIs from the natural logs of the reported RRs and CIs across categories of hemoglobin A1c by using the method of Greenland and Longnecker[6]. Summary RRs and 95% CIs for a 1% increment in hemoglobin A1c were pooled using a random effects model. RRs and 95%CI were calculated from the number of AF and cases in different HbA1c levels when them were not directly provided. All data were analyzed by Review Manager version 5.30 software (the Nordic Cochrane Center, Rigshospitalet, Denmark) and Stata software (Version 14.0, Stata Corp LP, College Station, Texas, USA). We performed the nonlinear dose-response analysis by using the robust error meta-regression method (REMR) described by Xu et al.[7] This method is based on a "one-stage approach" which treating each study as a cluster of the whole sample and considering the within study correlations by clustered robust error. It requires known levels of hemoglobin A1c and RRs with variance estimates for at least two quantitative exposure categories. For studies that did not set the highest hemoglobin A1c group as a reference, data were transformed using a method described by Hamling et al.[8] which requires the number of cases and participants in each category. If these data could not be obtained from an article, the evidence was not pooled. The category hemoglobin A1c concentration within each study for dose-response meta-analysis was calculated according to the methods of previous study. To assess the heterogeneity of RRs across studies, the I2 (95% CI) statistic was calculated with the following interpretation: low heterogeneity, defined as I2 < 50%; moderate heterogeneity, defined as I2 50% to 75%; and high heterogeneity, defined as I2 >75%[9]. If there was evidence of publication bias, we additionally applied trim and fill methods to adjust for publication bias. Sensitivity analyses excluding one study at a time were conducted to clarify whether the results were simply due to one large study or a study with an extreme result. A P value < 0.05 was considered statistically significant.

## Results

### Study selection

Our comprehensive retrieval identified 1,836 studies in our initial database search. After removing duplicates and studies with inadequate information on HbA1c and AF, 22 studies

were reviewed in more detail. After checking the full text, 8 were excluded for the following reasons: a) they were focused on recurrent AF (n = 1); b) they were reviews or case reports (n = 2); or c) they were investigating the association between glucose tolerance and AF (n = 4); d) they were using data from another study that was retrieved (n = 1). Finally, 14 studies were included in this meta-analysis [4–9, 13–15, 17–21] (**S1 Fig**).

## Study characteristics and quality

Detailed characteristics of the included studies are presented in **Table 1**. Fourteen studies [4–9, 14, 15, 17–21] with 17,914 AF cases among 352,325 participants were included in this meta-analysis. These studies were published between 2008 and 2017. The sample sizes of the included studies varied from 82 to 52,448. The mean age ranged from 53 to 72 years. Among the fourteen articles, six [4, 9, 13, 17, 19, 21] included diabetes or prediabetes patients, four [6–8, 18] included subjects without known diabetes and four [5, 14, 15, 20] studies reported both population with or without diabetes. Eight [4–9, 13, 18] studies examined the serum HbA1c status and AF, and 6[14, 15, 17, 19–21] focused on POAF in patients undergoing CABG.

We used the Newcastle-Ottawa Scale score to evaluate the quality of the included articles [22]. Only two studies were scored as low quality, with a NOS score of 5. The other twelve were scored as high quality (score ≥6) (**S3 Table**).

## Dose-response association between circulating HbA1c and incident AF

Eight studies [4, 5, 7–9, 11, 13, 18] with 16,516 cases among 344,709 participants were included in the dose-response analysis of HbA1c and AF. The summary RR for a 1% increase in HbA1c was 1.16 (95% CI: 1.07–1.27, $I^2$ = 52%, P = 0.0004) (**Fig 1**). Heterogeneity was reduced to 0% when excluding one study with short follow-up (acute myocardial infarction patients)[7], and the results were still significant (RR = 1.14, 95% CI: 1.09–1.19). In the sensitivity analysis conducted by omitting one study at a time, the pooled results were not significantly changed.

In addition, in the subgroup of patients, the results were still positive in diabetes or impaired glucose tolerance patients (RR = 1.13, 95% CI: 1.08–1.19, $I^2$ = 0%, P<0.0001) (**Fig 2**), with no evidence of heterogeneity. The result was still significant in subjects without known diabetes patients (RR = 1.12, 95% CI: 1.05–1.20, $I^2$ = 17%, P = 0.0009), with no evidence of heterogeneity.

Moreover, we conducted a nonlinear dose-response analysis to investigate the dose-specific effects between HbA1c and AF. Because of data restriction, the diabetes population was not included in this nonlinear model. We found a nonlinear ($P_{nonlinear}$ = 0.04) relationship between HbA1c and AF (**Fig 3**) in subjects without known diabetes. The curves showed that HbA1c over 6.3% significantly increased the risk of AF. The curves did not change significantly when the AMI patients were excluded.

## Association between HbA1c and POAF in patients undergoing CABG

Six[14, 15, 17, 19–21] studies that included 1,335 cases and 7,616 patients were included in this analysis. Serum HbA1c (per 1% increase) was not associated with POAF incidence (RR: 1.00, 95% CI: 0.84–1.20, $I^2$ = 79%, P = 0.96) (**Fig 4**). The results did not significantly change when the univariate analysis was excluded (RR: RR: 0.92, 95% CI: 0.80–1.06, $I^2$ = 72%, P = 0.27).

In the subgroup, serum HbA1c (per 1% increase) was not associated with the risk of AF in subjects with diabetes (RR: 1.08, 95% CI: 0.86–1.36, $I^2$ = 82%, P = 0.52) or without known diabetes (RR: 0.91, 95% CI: 0.73–1.12, P = 0.37) (**S2 Fig**). Nonlinear dose-response analysis was not possible because of limited information.

**Table 1. Basic characteristics of the 14 articles included in the meta-analysis.**

| Author, publication year, country | Population, Study design, follow-up duration | Source of participant, Cases/N | Definition of AF, Measurement H1Ac1, Mean age (years), male (%) | H1Ac1 expose level | RR (95% CI) | Adjustment for confounders |
|---|---|---|---|---|---|---|
| Halkos, 2008, USA [15] | CABG, Retrospective cohort, NA | Emory University Hospital, 549/3089 | ECG, NA, 63, 73 | Per 1% | 0.89 (0.80–0.98) | age, female, caucasian, renal failure, stroke, NYHA class IV,CCS class, main disease, No. diseased vessels, chronic lung disease, arrhythmia, peripheral vascular disease, perioperative factors. intraoperative glucose, postoperative glucose, arterial grafts, vein grafts, total grafts, CPB time, CPB used, LITA or BITA used |
| Matsuura, 2009, Japan [19] | CABG, Retrospective cohort, 2.1 years | Chiba University Hospital, 26/101 | NA, ion capture assay, 65, 79 | Per 1% | 0.87 (0.64–1.19)# | NA |
| Tsuruta, 2011, Japan [21] | CABG, Prospective cohort, 3.6 years | Juntendo University Hospital of Japan, 36/305 | NA, NA, 60, 79 | <6.5%; | Ref | NA |
| | | | | 6.5%-7.5% | 0.89 (0.36–2.21) | |
| | | | | ≥7.5% | 1.25 (0.58–2.71) | |
| | | | | Per 1%. | 1.12 (0.76–1.65) | |
| Kinoshita, 2012, Japan [14] | CABG, Case-control, NA | Shiga University of Medical Science, 159/805 | ECG, high-performance liquid chromatography, 68, 80 | 3.8–5.6% | Ref | age, sex, BMI, chronic kidney disease, chronic pulmonary disease, hypertension, triple vessel disease, ejection fraction of <40%, left atrial dimension, preoperative beta blockers, preoperative statins, inotropic support for >24 h, and transfusion |
| | | | | 6.8–11.4% | 0.55 (0.35–0.88) | |
| | | | | per 1% | 0.78 (0.63–0.95) | |
| Surer, 2016, Turkey [20] | CABG, Retrospective cohort, NA | Diskapi Yildirim Beyazit Training & Research Hospital, 12/72 | ECG, liquid chromatography, 63, 56 | Per 1% | 3.92 (1.92–7.99) | NA |
| Abbaszadeh, 2017, Iran [17] | CABG, Prospective cohort, 1 years | Tehran University of Medical Sciences, 109/708 | ECG, immunoturbidimetric, 61, 61 | Per 1% | 1.06 (0.93–1.2) | age, duration of diabetes, COPD, renal failure and hypertension, postoperative use of beta-blockers and calcium channel blockers, left atrial size, cardiopulmonary pump duration, and cross clamp time |
| Dublin, 2010, USA [6] | Population-based, Prospective case-control, 9.4 years | Group Health database of Research Institute of US, 1410/3613 | GH electronic data, calculated by total glycosylated hemoglobin, 70, 41 | 5%* | Ref | age, sex, calendar year, treated hypertension, and BMI |
| | | | | ≤7% | 1.06 (0.74–1.51) | |
| | | | | 7–8% | 1.48 (1.09–2.01) | |
| | | | | 8–9% | 1.46 (1.02–2.08) | |
| | | | | >9% | 1.96 (1.22–3.14) | |
| | | | | Per 1% | 1.14 (0.96–1.35) | |

*(Continued)*

**Table 1.** (Continued)

| Author, publication year, country | Population, Study design, follow-up duration | Source of participant, Cases/N | Definition of AF, Measurement H1Ac1, Mean age (years), male (%) | H1Ac1 expose level | RR (95% CI) | Adjustment for confounders |
|---|---|---|---|---|---|---|
| Huxley, 2012, USA [5] | DM or non-DM, Prospective cohort study, 14.5 years | ARIC study, 1311/13025 | ECG; high-performance liquid chromatography, 57, 44 | Per 1% (non-DM) | 1.05 (0.96–1.15) | age, study site, education, income, prevalent CHD, BMI, systolic BP, antihypertensive medications, and smoking |
| | | | | Per 1% (DM) | 1.13 (1.07–1.20) | |
| Iguchi, 2012, Japan [18] | Population-based, Prospective cohort, NA | Kurashiki city Public Health Center, 1161/52448 | ECG, NA, 72, 34 | Per 1% | 1.18 (1.09–1.28) | age, sex, BMI, chronic kidney disease, COPD, hypertension, triple vessel disease, ejection fraction of <40%, left atrial dimension, preoperative beta blockers, preoperative statins, inotropic support for >24 h, and transfusion |
| Turgut, 2013, Turkey [9] | DM, Retrospective Case-control, NA | Cumhuriyet University of Turkey, 81/162 | Medical records, turbidimetric immunoinhibition assay, 63, 52 | Per 1% | 1.87 (0.747–3.014) | age, male gender, HT, smoking, history of CAD, previous CVA, microalbuminuria, retinopathy, duration of diabetes, BMI, total cholesterol, LDL cholesterol, HDL cholesterol, triglyceride, platelet count, and MPV |
| Latini,2013, Italy [4] | Population with impaired glucose tolerance, Post hoc analysis of RCT, 6.5 years | NAVIGATOR study, 613/8943 | ECG or investigator-reported event, NA, 63, 49 | Per 1% | 1.11 (0.91–1.32) | age, height, abnormal ECG, heart rate, COPD, BP, weight, hypertension, eGFR, fasting plasma glucose, CHF, race, CHD outcomes, platelet |
| Sandhu, 2014, Canada [8] | Population-based (women), Prospective cohort, 16.4 years | Women's Health Study, 1039/34720 | ECG or medical report, NA, 53, 0 | Paroxysmal AF | | interim MI, stroke, revascularization, and HF |
| | | | | Per 1% | 1.08 (0.95–1.22)[#] | |
| | | | | ≤4.84% | Ref | |
| | | | | 4.84–5.00% | 0.90 (0.69–1.17) | |
| | | | | 5.00–5.19% | 0.99 (0.77–1.27) | |
| | | | | >5.19% | 0.76 (0.58–1.00) | |
| | | | | Nonparoxysmal AF | | |
| | | | | ≤4.84% | Ref | |
| | | | | 4.84–5.00% | 1.30 (0.85–1.97) | |
| | | | | 5.00–5.19% | 1.43 (0.96–2.15) | |
| | | | | >5.19% | 1.48 (0.98–2.22) | |

*(Continued)*

**Table 1.** (Continued)

| Author, publication year, country | Population, Study design, follow-up duration | Source of participant, Cases/N | Definition of AF, Measurement H1Ac1, Mean age (years), male (%) | H1Ac1 expose level | RR (95% CI) | Adjustment for confounders |
|---|---|---|---|---|---|---|
| Blasco, 2014, Spain [7] | AMI, Prospective cohort, 8 days | University Clinic Hospital of Valencia, 12/601 | ECG, ion-exchange high-performance liquid chromatography, 62, 78 | <5.5% | Ref | age, sex, classical cardiovascular risk factors, Killip class, history of previous MI, serum creatinine, serum glycaemia, systolic, BP and heart rate on admission. |
| | | | | 5.5–6.4% | 1.68 (0.64–4.39) | |
| | | | | >6.4% | 29.74 (10.79–81.94) | |
| | | | | per 1% | 3.24 (2.41–4.35)# | |
| Dahlqvist, 2017, Sweden [13] | type 1 diabetes, prospective case-control study | Swedish National Diabetes Registry, 1283/ 216852 | ECG, 35.4, 55 | Control | Ref | age, sex, education level, birthplace, country of birth, diabetes duration, level of education, mean systolic BP, BMI, HDL cholesterol, and LDL cholesterol, and time-updated smoking, use of blood pressure-lowering medication, use of lipid-lowering drugs, and insulin delivery method, CAD, HF, valve disease, stroke, and cancer |
| | | | | <6.9 | 1.03 (0.84–1.27) | |
| | | | | 7.0–7.8 | 1.11 (0.97–1.28) | |
| | | | | 7.8–8.7 | 1.28 (1.12–1.47) | |
| | | | | 8.8–9.6 | 1.4 (1.15–1.69) | |
| | | | | >9.7 | 2.32 (1.8–2.99) | |
| | | | | Per 1% | 1.14 (1.06–1.24) | |

Abbreviations: CABG = coronary artery bypass grafting; DM = diabetes mellitus; AMI = acute myocardial infarction; ARIC = The Atherosclerosis Risk in Communities study; H1Ac1 = Hemoglobin A1c; ECG = electrocardiogram; AF = atrial fibrillation; COPD = chronic obstructive pulmonary disease. NYHA = New York Heart Association; BMI = Body Mass Index HDL = high density lipoprotein; CPB = cardiopulmonary bypass; CCS = Canadian Cardiovascular Society; LITA BITA = bilateral internal thoracic artery; eGFR = estimated creatinine clearance; CHD = Coronary heart disease; HF = heart failure; BP = blood pressure; CAD = coronary artery disease; NAVIGATOR = Nateglinide and Valsartan in Impaired Glucose Tolerance Outcomes Research. NA = not available.

* H1Ac1 level of no history of DM

#Calculated from the number of AF occurrence in different HbA1c levels

### Publication bias

The funnel plots for AF (N = 8) and POAF (N = 6) were not evaluated due to the limited number of studies (N<10), according to the guideline [23].

### Discussion

To the best of our knowledge, this is the first meta-analysis to evaluate the dose-response association between HbA1c and AF. Our results showed evidence of a positive nonlinear dose-response association between HbA1c and AF. For every 1% increase in HbA1c level, the risk of AF increased by 28%. The association was positive in diabetes patients and subjects without diabetes or with unknown diabetes status. Interestingly, unlike in previous reports, elevated serum HbA1c level was not a protective factor for AF in patients after CABG, whether in diabetes or in nondiabetes.

| Study or Subgroup | log[Risk Ratio] | SE | Weight | Risk Ratio IV, Random, 95% CI | Risk Ratio IV, Random, 95% CI |
|---|---|---|---|---|---|
| Blasco et al, 2014 | 1.1754184 | 0.316918 | 1.7% | 3.24 [1.74, 6.03] | |
| Dahlqvist et al,2017 | 0.1397619 | 0.0590464 | 19.8% | 1.15 [1.02, 1.29] | |
| Dublin et al,2010 | 0.131028 | 0.086971 | 13.7% | 1.14 [0.96, 1.35] | |
| Huxley et al, 2012 | 0.09531 | 0.037171 | 25.7% | 1.10 [1.02, 1.18] | |
| Iguchi et al, 2012 | 0.165514 | 0.04099 | 24.7% | 1.18 [1.09, 1.28] | |
| Latini et al, 2013 | 0.09531 | 0.094883 | 12.3% | 1.10 [0.91, 1.32] | |
| Sandhu et al, 2014 | 0.09531 | 0.528005 | 0.6% | 1.10 [0.39, 3.10] | |
| Turgut et al, 2013 | 0.625938 | 0.355857 | 1.4% | 1.87 [0.93, 3.76] | |
| | | | | | |
| Total (95% CI) | | | 100.0% | 1.16 [1.07, 1.27] | |

Heterogeneity: Tau² = 0.01; Chi² = 14.56, df = 7 (P = 0.04); I² = 52%
Test for overall effect: Z = 3.57 (P = 0.0004)

**Fig 1. Forest plot of the association between HbA1c level and risk of atrial fibrillation, per 1% increase in HbA1c.**

A previous meta-analysis reported a weak association between HbA1c and the risk of AF in prospective studies; however, no positive association was observed in the primary analysis[24]. The inconsistent results between the previous study and ours might have several reasons. First, diabetes patients and subjects without known diabetes were combined for analysis. The relationship between HbA1c and AF in diabetes patients and subjects without known diabetes is still unknown. Another problem is that the general population and CABG patients were also combined for analysis in their primary results. This might not be appropriate because patients with CABG have great heterogeneity compared with the general population. Third, not all of

| Study or Subgroup | log[Risk Ratio] | SE | Weight | Risk Ratio IV, Random, 95% CI | Risk Ratio IV, Random, 95% CI |
|---|---|---|---|---|---|
| **2.2.1 DM** | | | | | |
| Dahlqvist et al,2017 | 0.1397619 | 0.0590464 | 18.2% | 1.15 [1.02, 1.29] | |
| Huxley et al(DM), 2012 | 0.122218 | 0.029251 | 74.2% | 1.13 [1.07, 1.20] | |
| Latini et al, 2013 | 0.09531 | 0.094883 | 7.1% | 1.10 [0.91, 1.32] | |
| Turgut et al, 2013 | 0.625938 | 0.355857 | 0.5% | 1.87 [0.93, 3.76] | |
| Subtotal (95% CI) | | | 100.0% | 1.13 [1.08, 1.19] | |

Heterogeneity: Tau² = 0.00; Chi² = 2.15, df = 3 (P = 0.54); I² = 0%
Test for overall effect: Z = 5.00 (P < 0.00001)

| | | | | | |
|---|---|---|---|---|---|
| **2.2.2 Non-DM** | | | | | |
| Dublin et al,2010 | 0.131028 | 0.086971 | 14.0% | 1.14 [0.96, 1.35] | |
| Huxley(No DM) et al, 2012 | 0.04879 | 0.046067 | 39.4% | 1.05 [0.96, 1.15] | |
| Iguchi et al, 2012 | 0.165514 | 0.04099 | 46.2% | 1.18 [1.09, 1.28] | |
| Sandhu et al, 2014 | 0.09531 | 0.528005 | 0.4% | 1.10 [0.39, 3.10] | |
| Subtotal (95% CI) | | | 100.0% | 1.12 [1.05, 1.20] | |

Heterogeneity: Tau² = 0.00; Chi² = 3.62, df = 3 (P = 0.31); I² = 17%
Test for overall effect: Z = 3.33 (P = 0.0009)

**Fig 2. Forest plot of HbA1c and atrial fibrillation incidence among diabetes or unknown diabetes, per 1% increase in HbA1c.**

**Fig 3. HbA1c and risk of atrial fibrillation in patients with unknown diabetes, nonlinear dose-response analysis.** The solid line and the dashed lines represent the estimated relative risk and the 95% confidence interval, respectively.

the available studies could not be pooled because of the types of methods used in their study (semi-quantitative). More importantly, only pooling category-specific relative risks and their confidence intervals (e.g. highest vs lowest) leading to a bias in statistics. In contrast to the previous meta-analysis, our results suggest that increased serum HbA1c levels were associated

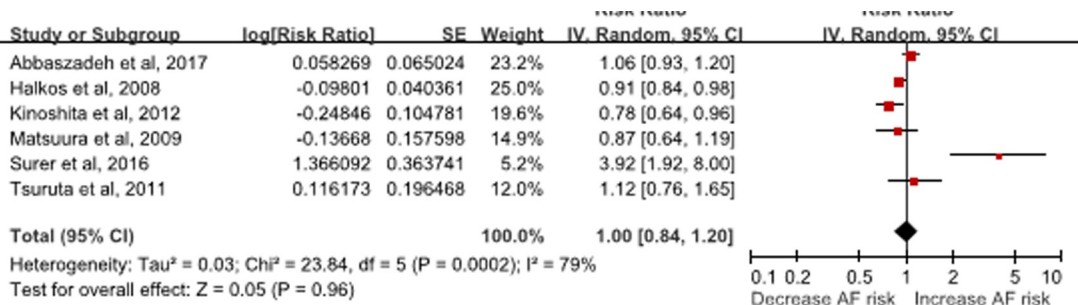

| Study or Subgroup | log[Risk Ratio] | SE | Weight | Risk Ratio IV. Random, 95% CI | Risk Ratio IV. Random, 95% CI |
|---|---|---|---|---|---|
| Abbaszadeh et al, 2017 | 0.058269 | 0.065024 | 23.2% | 1.06 [0.93, 1.20] | |
| Halkos et al, 2008 | -0.09801 | 0.040361 | 25.0% | 0.91 [0.84, 0.98] | |
| Kinoshita et al, 2012 | -0.24846 | 0.104781 | 19.6% | 0.78 [0.64, 0.96] | |
| Matsuura et al, 2009 | -0.13668 | 0.157598 | 14.9% | 0.87 [0.64, 1.19] | |
| Surer et al, 2016 | 1.366092 | 0.363741 | 5.2% | 3.92 [1.92, 8.00] | |
| Tsuruta et al, 2011 | 0.116173 | 0.196468 | 12.0% | 1.12 [0.76, 1.65] | |
| | | | | | |
| Total (95% CI) | | | 100.0% | 1.00 [0.84, 1.20] | |

Heterogeneity: Tau² = 0.03; Chi² = 23.84, df = 5 (P = 0.0002); I² = 79%
Test for overall effect: Z = 0.05 (P = 0.96)

**Fig 4. Forest plot of HbA1c and atrial fibrillation incidence in patients undergoing coronary artery bypass, per 1% increase in HbA1c.**

with an increased risk of AF in diabetes patients using the dose-response method. In the linear model, the summary RRs for per 1% increase in HAb1Ac were positive, both in patients with or without known diabetes. Moreover, in the non-linear model, we found there is a nonlinear ($P_{nonlinear} = 0.04$) relationship between HbA1c and AF in patients without known diabetes. The dose-specific curve showed that HbA1c over 6.3% significantly increased the risk of AF, and rose steeply thereafter. These results are consistent with an article that found a significant positive correlation between HbA1c levels and the duration of AF (r = 0.408, p = 0.005) and worse cardiovascular outcomes[25], thereby suggesting the important role of abnormal glucose metabolism in increasing AF burden. Several underlying pathophysiological mechanisms might explain this association. First, long-term high levels of HbA1c could induce the production of advanced glycation end products (AGEs) and increase the level of tissue growth factor, causing atrial structural remodeling and atrial enlargement[26]. On the other hand, it has been proposed that AGEs, including HbA1c, may play a crucial role in atrial remodeling via the induction of reactive oxygen species and may eventually cause electrolyte imbalance and systemic inflammation[27]. All of the above effects may lead to greater AF vulnerability.

Moreover, our study extended the association between HbA1c and AF risk. Our dose-response curves showed that HbA1c>6.3% significantly increased the risk of AF in subjects without diabetes or with unknown diabetes status. Considering the close association between hyperglycemic state and AF, this result was not surprising. Hyperglycemia not only could prolong the atrial conduction times and P wave dispersion in prediabetic patients but also was an important risk factor for AF even in healthy populations without comorbidities [28, 29]. In fact, several meta-analyses also showed that HbA1c>6.3% significantly increased the cardiovascular mortality and risk of events in the diabetic population [30]. Thus, our findings reinforce the idea that HbA1c is a potential factor in algorithms to calculate cardiovascular risk in clinical settings[31]. We also noted that a study based type 1 diabetes population showed that the threshold of HbA1c level increasing the AF risk was 7.8%-8.7%[13]. These results suggested that in people with untreated diabetes, the cardiovascular risk (such as AF) may be higher compared with known patients with diabetes, which might emphasize the importance of screening and treatment for diabetes.

We did not find a protective role of HbA1c in POAF, in both individuals with diabetes and without diabetes or with unknown diabetes status. However, we should interpret these results with caution. First, the number of including studies was limited. Only one study examined the association between HbA1c and POAF in nondiabetic patients. Second, there was significant heterogeneity in the results of POAF. Because of a limited number of included studies (N<10), meta-regression was not performed, according to the guideline [23]. Therefore, the source of heterogeneity was not identified, and it may have come from the study design or baseline characteristics of the patients. Furthermore, it is notable that only half (N = 3)[14] of the included studies adjusted for potential clinical covariates in their results, and the other studies did not analyze the association between HbA1c and AF in multivariate regression analyses. Many clinical covariates and many important clinical confounding factors (e.g., anti-arrhythmia drugs) were not fully adjusted for, which might have influenced the risk of POAF. In addition, the sample size in the majority of included studies was relatively small. Thus, larger, well-designed studies are needed to explore whether there was a protective effect of preoperative HbA1c on POAF in patients undergoing CABG.

## Study limitations

The present meta-analysis has several limitations. First, although majority of studies adjusted the potential confounding (e.g. age, sex, BMI), this was a meta-analysis of observational

studies, which cannot directly prove causation, and the unmeasured and insufficiently measured variables could have resulted in residual confounding. Second, there was substantial heterogeneity among our primary and subgroup analyses of POAF, which may have come from different study designs and patient baseline values. Third, the HbA1c level was measured only once, at baseline, in all of the included studies. Thus, regression dilution bias may have been induced in our results because of the within-subject variation in HbA1c level and measurement error.

## Conclusion

Our dose-response results suggest that serum higher HbA1c was associated with an increased risk of AF, both in diabetes and without diabetes or with unknown diabetes status. However, no association was found between HbA1c and POAF in patients after CABG. Undoubtedly, because of the limited sample size and significant heterogeneity among the included studies, these conclusions should be interpreted with caution, and further well-designed studies with larger sample sizes are needed to explore the association between serum HbA1c level and the risk of POAF.

## Supporting information

**S1 Fig. Flowchart of study selection.**
(DOCX)

**S2 Fig. Forest plot of HbA1c and atrial fibrillation incidence among diabetes and non-diabetes patients undergoing coronary artery bypass, per 1% increase.**
(DOCX)

**S1 Table.**
(DOCX)

**S2 Table. Search strategy.**
(DOCX)

**S3 Table. Quality assessment of included studies.**
(DOCX)

## Author Contributions

**Conceptualization:** Huilei Zhao, Kaibo Mei.

**Data curation:** Huilei Zhao, Menglu Liu, Zhifeng Chen, Kaibo Mei, Peng Yu.

**Formal analysis:** Huilei Zhao, Menglu Liu.

**Funding acquisition:** Lixia Xie.

**Investigation:** Menglu Liu, Kaibo Mei.

**Methodology:** Menglu Liu, Zhifeng Chen, Lixia Xie.

**Project administration:** Huilei Zhao.

**Resources:** Huilei Zhao.

**Software:** Kaibo Mei, Lixia Xie.

**Visualization:** Huilei Zhao.

**Writing – original draft:** Huilei Zhao, Lixia Xie.

**Writing – review & editing:** Peng Yu, Lixia Xie.

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
