## [Decision Letter · Decision Letter 0]

4 Nov 2019

PONE-D-19-25933

The Relationship between Hemoglobin A1c and Risk of Atrial Fibrillation in Patients with and without Known Diabetes: Evidence from a Dose-response Meta-analysis

PLOS ONE

Dear authors,

Thank you for submitting your manuscript to PLOS ONE. After careful consideration, we feel that it has merit but does not fully meet PLOS ONE’s publication criteria as it currently stands. Therefore, we invite you to submit a revised version of the manuscript that addresses the points raised during the review process.

We would appreciate receiving your revised manuscript by 30th December 2019. To enhance the reproducibility of your results, we recommend that if applicable you deposit your laboratory protocols in protocols.io, where a protocol can be assigned its own identifier (DOI) such that it can be cited independently in the future. For instructions see: http://journals.plos.org/plosone/s/submission-guidelines#loc-laboratory-protocols

We look forward to receiving your revised manuscript.

Kind regards,

Ronpichai Chokesuwattanaskul, M.D.

Academic Editor

PLOS ONE

Journal Requirements:

2. Please move your methodologies from the supplementary files to the main text.

Additional Editor Comments (if provided):

Reviewers' comments:

Reviewer's Responses to Questions

**Comments to the Author**

1. Is the manuscript technically sound, and do the data support the conclusions?

Reviewer #1: Yes

Reviewer #2: Partly

Reviewer #3: Partly

Reviewer #4: Partly

Reviewer #5: Yes

2. Has the statistical analysis been performed appropriately and rigorously? 

Reviewer #1: I Don't Know

Reviewer #2: I Don't Know

Reviewer #3: No

Reviewer #4: No

Reviewer #5: Yes

3. Have the authors made all data underlying the findings in their manuscript fully available?

Reviewer #1: Yes

Reviewer #2: No

Reviewer #3: No

Reviewer #4: Yes

Reviewer #5: Yes

4. Is the manuscript presented in an intelligible fashion and written in standard English?

Reviewer #1: Yes

Reviewer #2: Yes

Reviewer #3: Yes

Reviewer #4: Yes

Reviewer #5: Yes

5. Review Comments to the Author

Reviewer #1: This manuscript reported in form of meta-analysis that higher HbA1c was associated with an increased risk of AF, both in diabetes and in without diabetes or with unknown diabetes. However, no association was found between HbA1c and POAF in patients undergoing CABG.

The issue has been reported for several times, for example very recently, a meta-analysis by Bohne el al. published on Front Physiol. 2019 Feb 26;10:135. doi: 10.3389/fphys.2019.00135. eCollection 2019.

Minor revision:

1. I think this is not the first meta-analysis to evaluate the dose-response association between HbA1c and AF. Please discuss about what is the difference between the manuscript and previous meta-analysis.

Reviewer #2: The authors of this study did not provide us with the following:-

. I could not find any data on the serach startgey or the inclusion and exclusion criteria

. There are many confuncting factors that could explaint the increased risk of A.fib , it is not clear what confunding factors were corrected for in this study. If none, this should be stated clearly.

. I could not see any statisticain in the list of the authors. This study has to be reviewed by a statisticain to ensure that the results are robust.

Reviewer #3: 1- The authors need to describe the rationale for the review in the context of what is already known and the aim of the study in a single clear sentence.

2- Mention the search terms were used to identify relevant published articles

3- The starting time for searching should be mentioned.

4- There is needed a flow diagram to show study selection.

5- The authors should present results of any assessment of risk of bias across studies.

6- Data extraction should be described more.

7- There any additional analysis? It should be mentioned.

8- Did RCTs and observational studies analyze all together?

9- Statistical analysis should be written in details.

Reviewer #4: In the protocol, authors were mentioned that observation studies included in the meta-analysis, but in article RCT and Observation studies included in the meta-analysis. Please, authors, descript this different.

In the cohort studies results in figure 3 showed. Please explain this results in meta-analysis and present reasons to conducted meta-analysis for detecting this relationship.

In the method section way to extraction data not clear.

The quality assessment and ROB for observation studies conducted by the NOS checklist. Risk of bias for RCT not mentioned. Please clarify.

The meta-regression not conducted.

Authors were mentioned that HbA1c over 6.3% significantly increased the risk of AF. This result seems incorrect because authors use nonlinear analysis, so authors cannot do a linear interpretation.

Reviewer #5: Interesting paper.

I think that the most releant limitation is represented by absence of relationship (at least declared) at multivariate analysis.

SO the authors should

1) abstract if possible independendent HR/OR from the studies

2) pool with random effect together

because it is not clear if these HR/OR were abstracted with independent or not random/fixed effects

6. PLOS authors have the option to publish the peer review history of their article (what does this mean?). If published, this will include your full peer review and any attached files.

Reviewer #1: No

Reviewer #2: Yes: Mohammed Bashir

Reviewer #3: No

Reviewer #4: No

Reviewer #5: Yes: Fabrizio D'Ascenzo

---

## [Author Response · Author response to Decision Letter 0]

18 Nov 2019

Response to the comments of the reviewers and revisions

My co-authors and I thank you sincerely for the opportunity to revise our manuscript. Of note, we found a recent prospective case-control study (Dahlqvist et al)(1) were not included in our current study. We apologized for this mistakes. Thus, we included this study and re-performed the meta-analysis. With this adjustment, although the specific numerical values may have changed, the corresponding results did not significantly change in the updated manuscript. We again thank the reviewers for raising those points that helped us to improve the quality and impact of our study. Point-by-point responses to the editors and reviewers are listed as followed.

1. Dahlqvist S, Rosengren A, Gudbjörnsdottir S, et al. Risk of atrial fibrillation in people with type 1 diabetes compared with matched controls from the general population: a prospective case-control study. The Lancet Diabetes & Endocrinology. 2017; 5; 799-807.

Reviewer #1: This manuscript reported in form of meta-analysis that higher HbA1c was associated with an increased risk of AF, both in diabetes and in without diabetes or with unknown diabetes. However, no association was found between HbA1c and POAF in patients undergoing CABG.

The issue has been reported for several times, for example very recently, a meta-analysis by Bohne et al. published on Front Physiol. 2019 Feb 26;10:135. doi: 10.3389/fphys.2019.00135. eCollection 2019.

Response: Thanks for your helpful comments. Although Bohne et al. performed a meta-analysis about the association between diabetes and AF, the relationship between HbA1c and AF was not investigated. Actually, the association between HbA1C and AF listed in Figure 2 of their studies was reproduced from the original research of Dahlqvist et al. Therefore, the exposure-effect of HA1bc on AF has not been examined in the study of Bohne et al.

Minor revision:

Comment: 1. I think this is not the first meta-analysis to evaluate the dose-response association between HbA1c and AF. Please discuss about what is the difference between the manuscript and previous meta-analysis.

Response: Thanks for your helpful comments. Although the previous study has evaluated HbA1c and AF(1). However, they had several limitations. First, they only use a standard approach, HbA1c levels were analyzed as either a categorical (highest vs lowest) or continuous variable, which resulting in pooling all the evidence was not available. More importantly, only pooling category-specific relative risks and their confidence intervals (e.g. highest vs lowest) leading to a bias in statistics. Secondly, the AF and POAF post-CABG were analyzed combined, which was not appropriated. Third, they did analyze known diabetes or unknown diabetes separately. However, there was large heterogeneity in the two groups of populations. By using novel methods, we first examined the dose-response relationship between HbA1c and AF. In the linear model, the summary RRs for per 1% increase in HAb1Ac were positive, both in patients with or without known diabetes. Moreover, in the non-linear model, we found there is a nonlinear (Pnonlinear=0.04) relationship between HbA1c and AF in patients without known diabetes. The dose-specific curve showed that HbA1c over 6.3% significantly increased the risk of AF, and rose steeply thereafter. This result was consistent with another dose-response meta-analysis that reporting higher HbA1c level higher HbA1c level is associated with increased mortality from all causes and CVD among subjects without known diabetes(2). We have added this point to the discussion part.

1. Qi W, Zhang N, Korantzopoulos P, et al. Serum glycated hemoglobin level as a predictor of atrial fibrillation: A systematic review with meta-analysis and meta-regression. PLoS One. 2017; 12; e0170955.

2. Zhong GC, Ye MX, Cheng JH, Zhao Y, Gong JP. HbA1c and Risks of All-Cause and Cause-Specific Death in Subjects without Known Diabetes: A Dose-Response Meta-Analysis of Prospective Cohort Studies. Sci Rep. 2016; 6; 24071.

Reviewer #2: The authors of this study did not provide us with the following:-

Comment: I could not find any data on the search strategy or the inclusion and exclusion criteria

Response: Thanks for your helpful comments. The search strategy, inclusion, and exclusion criteria were reported in detail in the supplement method of the previous manuscript. We have moved these parts to the text in the revised manuscript. (line 52-63, page 8-9, marked in red).

Comment: There are many confounding factors that could explain the increased risk of A.fib , it is not clear what confounding factors were corrected for in this study. If none, this should be stated clearly.

Response: Thanks for your helpful comments. We added the adjustments of each study in Table 1. All the studies reporting the AF were adjusted for confounding factors (e.g. age, sex, BMI). However, as discussed in line 169, page 17, for the outcomes for POAF, only half (N=3) of the included studies adjusted for potential clinical covariates in their results. Many clinical covariates and many important clinical confounding factors (e.g., anti-arrhythmia drugs) were not fully adjusted, which might have influenced the risk of POAF. In addition, the sample size in the majority of included studies was relatively small. Thus, larger, well-designed studies are needed to explore whether there was a protective effect of preoperative HbA1c on POAF in patients undergoing CABG. 

Comment: I could not see any statistician in the list of the authors. This study has to be reviewed by a statistician to ensure that the results are robust.

Response: Thanks for your helpful comments. Our manuscript received a statistical review (Zhifeng Chen, a statistician [School of Public Health, University of International Studies and Trade, Fuzhou, China]). 

Reviewer #3: 

Comment: 1- The authors need to describe the rationale for the review in the context of what is already known and the aim of the study in a single clear sentence.

Response: Thanks for your helpful comments. The association between HbA1c and AF has been investigated in previous studies. However, the evidence is inconsistent. Thereafter, although a meta-analysis showed that levels of HbA1c increased the risk of AF. However, with several limitations. First, they only use a standard approach, HbA1c levels were analyzed as either a categorical (highest vs lowest) or continuous variable, which resulting in pooling all the evidence was not available. More importantly, only pooling category-specific relative risks and their confidence intervals (e.g. highest vs lowest) leading to a bias in statistics. Secondly, the AF and POAF post-CABG were analyzed combined, which was not appropriated. Third, they did analyze known diabetes or unknown diabetes separately. However, there was large heterogeneity in the two groups of populations. Moreover, several new studies with large populations were published, still with inconsistent results. Given this background, we used a novel dose-response meta-analysis to quantitatively investigate the dose-specific relationship between HbA1c and AF and whether serum HbA1c could reduce post-operation AF (POAF) after CABG. We have revised this issue in the revised manuscript. (line 45-52, page 5-6 and line137-147, page 15-16, marked in red)

Comment: 2- Mention the search terms were used to identify relevant published articles

3- The starting time for searching should be mentioned.

4- There is needed a flow diagram to show study selection.

Response: We apologized for your confusion. The search terms, starting time for searching and flow diagram was provided in the supplementary method or figures. We have moved the supplemental methods in the text in the revised manuscript. Please find the starting time for searching in line 53, page 7. The search strategy (including search terms) or flow diagram was shown in supplemental Table s2 or Figure S1, respectively.

Comment: 5- The authors should present the results of any assessment of the risk of bias across studies.

Response: Thanks for your kind comments. According to the guideline (Cochrane Handbook for Systematic Reviews, chapter10.4.3.1), publication bias is difficult to evaluate among reviews of 10 or fewer. Therefore, the publication bias of AF (n=8) or POAF (n=6) in the current study was not conducted as limited studies (N<10). This point was described in the methods section. (line 132, page 9, mark in red)

Comment: 6- Data extraction should be described more.

Response: We apologized for your confusion. Please find this issue in line 71-75, page 9. (marked in red)

Comment: 7- There any additional analysis? It should be mentioned.

Response: Thanks for your comments. First, in the primary analysis, although there is a moderate heterogeneity. However, the heterogeneity was reduced to 0% when excluding one study with short follow-up (acute myocardial infarction patients), which suggests the heterogeneity might come from a different population. Moreover, we performed a sensitivity analysis conducted by omitting one study at a time, the pooled results were not significantly changed. (line 115, page 13, marked in red) These analyses suggested our results were robust and stable. Therefore, we did not perform an additional subgroup analysis (exclude the subgroup of with or without diabetes) to find the source of heterogeneity. 

Comment: 8- Did RCTs and observational studies analyze all together?

Response: Thanks for your helpful comments. In fact, the RCTs and cohort studies cannot be pooled together in most cases. However, post hoc analysis of RCTs has been shown to be equivalent to observational studies and thus might allow RCTs to be pooled with observational studies1,2. In addition, exclusion of the post-hoc of RCT (Latin et al) did change the results (RR:1.23, P=0.004).

1. Hulot JS, Collet JP, Silvain J, Pena A, Bellemain-Appaix A, Barthelemy O, et al. Cardiovascular risk in clopidogrel-treated patients according to cytochrome P450 2C19*2 loss-of-function allele or proton pump inhibitor coadministration: a systematic meta-analysis. J Am Coll Cardiol. 2010;56(2):134-143.

2. Xu T, Huang Y, Zhou H, Bai Y, Huang X, Hu Y, et al. beta-blockers and risk of all-cause mortality in patients with chronic heart failure and atrial fibrillation-a meta-analysis. BMC Cardiovasc Disord. 2019;19(1):135.

Comment: 9- Statistical analysis should be written in detail.

Response: Thanks for your helpful comments. The statistical analysis was reported in detail in the supplement method previously manuscript. Now the detailed description of statistical analysis was provided in the revised manuscript. (line 75-90, page 10-11, marked in red)

Reviewer #4: 

Comment: In the protocol, authors were mentioned that observational studies included in the meta-analysis, but in article RCT and Observation studies included in the meta-analysis. Please, authors, descript this different.

Response: Thanks for your helpful comments. In fact, the RCTs and cohort studies cannot be pooled together in most cases. However, post hoc analysis of RCTs has been shown to be equivalent to observational studies and thus might allow RCTs to be pooled with observational studies (1,2). In addition, exclusion of the post-hoc of RCT (Latin et al) did change the results (RR:1.23, P=0.004).

1. Hulot JS, Collet JP, Silvain J, Pena A, Bellemain-Appaix A, Barthelemy O, et al. Cardiovascular risk in clopidogrel-treated patients according to cytochrome P450 2C19*2 loss-of-function allele or proton pump inhibitor coadministration: a systematic meta-analysis. J Am Coll Cardiol. 2010;56(2):134-143.

2. Xu T, Huang Y, Zhou H, Bai Y, Huang X, Hu Y, et al. beta-blockers and risk of all-cause mortality in patients with chronic heart failure and atrial fibrillation-a meta-analysis. BMC Cardiovasc Disord. 2019;19(1):135.

Comment: In the cohort studies results in figure 3 showed. Please explain these results in meta-analysis and present reasons to conducted meta-analysis for detecting this relationship.

Response: As data restriction, we only performed a non-linear analysis to examine the dose-specific relationship between HbA1c and AF in patients with without known diabetes. We found there is a nonlinear (Pnonlinear=0.04) relationship between HbA1c and AF in patients without known diabetes. The dose-specific curve showed that HbA1c over 6.3% significantly increased the risk of AF, and rose steeply thereafter. This result was consistent with another dose-response meta-analysis that reporting higher HbA1c level higher HbA1c level is associated with increased mortality from all causes and CVD among subjects without known diabetes. Of note, a studies based diabetes population showed that the threshold of HbA1c level increased AF was 7.8%-8.7%(1). These results suggested that in people with untreated diabetes, the cardiovascular risk (such as AF) may be higher compared with known patients with diabetes, which might emphasize the importance of screening and treatment for diabetes. We had explained the results of figure 3 in line 121-123, page 14 and line 160-163, page16-17.

1. Dahlqvist S, Rosengren A, Gudbjörnsdottir S, et al. Risk of atrial fibrillation in people with type 1 diabetes compared with matched controls from the general population: a prospective case-control study. The Lancet Diabetes & Endocrinology. 2017; 5; 799-807.

Comment: In the method section way to extraction data not clear.

Response: We apologized for your confusion. The statistical analysis was reported detail in the supplement methods previously manuscript. We have moved the supplemental methods in the text in the revised manuscript. Now the detail of statistical analysis was described in the revised manuscript. (line 75-90, page10-11, marked in red)

Comment: The quality assessment and ROB for observation studies conducted by the NOS checklist. Risk of bias for RCT not mentioned. Please clarify.

Response: We apologized for your confusion. As above mentioned, we treat post hoc analysis of RCTs as cohort studies. Thus, the use of the NOS scale to achieve quality assessment for hoc analysis of RCTs. We have added this point to the method section. (line 77, page 9, marked in red)

Comment: The meta-regression not conducted.

Response: Thanks for your helpful comments. We did not perform the meta-regression in the analysis of AF (n=6) or POAF(n=3) because of limited studies (N<10). According to the guideline, meta-regression was not recommended when numbers of studies were less than 10. We have added this point to the discussion. (line 156, page 17, marked in red)

1. Higgins JP, Green S. Cochrane Handbook for Systematic Reviews of Interventions Version 5.1.0 [updated March 2011]The Cochrane Collaboration, 2011. Available from http://handbook.cochrane.org.chanpter.

Comment: Authors were mentioned that HbA1c over 6.3% significantly increased the risk of AF. This result seems incorrect because authors use nonlinear analysis, so authors cannot do a linear interpretation.

Response: Thanks for your helpful comments. The sentence of “HbA1c over 6.3% significantly increased the risk of AF” was derived from the non-linear model. A non-linear model can provide the dose-specific information between the HbA1c level and the risk of AF. In the non-linear curve, we can see the when HbA1c over 6.3%, the RR was elevated and became statistically significant. The linear model was used to detect the “average effect”, which can't provide the threshold value of HbA1c of increasing the incidence of AF. 

Reviewer #5: Interesting paper.

Comment: I think that the most relevant limitation is represented by absence of relationship (at least declared) at multivariate analysis.

Comment: 1) abstract if possible independendent HR/OR from the studies

Comment: 2) pool with random effect together

because it is not clear if these HR/OR were abstracted with independent or not random/fixed effects

Response: We apologized for your confusion. We used a random-effects model in the current study.

We have added the adjustments of each study in table 1. Actually, all the data of studies reporting AF was extracted from multivariate analysis. Moreover, although there was a moderate Heterogeneity in the pooled RR for a 1% increase in HbA1c (RR: 1.14, 95% CI: 1.06–1.24). Heterogeneity was reduced to 0% when excluding one study with short follow-up (acute myocardial infarction patients), and the results were still significant (RR=1.13, 95% CI: 1.08-1.18), which suggested our result was robust. (line 110, page 13, marked in red)

However, for the outcomes for POAF, as discussed in the line 167, page 16, only half (N=3) of the included studies adjusted for potential clinical covariates in their results. Many clinical covariates and many important clinical confounding factors (e.g., anti-arrhythmia drugs) were not fully adjusted, which might have influenced the risk of POAF. In addition, the sample size in the majority of included studies was relatively small. Thus, larger, well-designed studies are needed to explore whether there was a protective effect of preoperative HbA1c on POAF in patients undergoing CABG. We have added this point to the discussion. (line 166-173, page 17, marked in red)

---

## [Decision Letter · Decision Letter 1]

17 Dec 2019

Dose-response analysis between hemoglobin A1c and risk of atrial fibrillation in patients with and without known diabetes

PONE-D-19-25933R1

Dear Authors,

We are pleased to inform you that your manuscript has been judged scientifically suitable for publication and will be formally accepted for publication once it complies with all outstanding technical requirements.

With kind regards,

Ronpichai Chokesuwattanaskul, M.D.

Academic Editor

PLOS ONE

Additional Editor Comments (optional):

Reviewer's Responses to Questions

**Comments to the Author**

1. If the authors have adequately addressed your comments raised in a previous round of review and you feel that this manuscript is now acceptable for publication, you may indicate that here to bypass the “Comments to the Author” section, enter your conflict of interest statement in the “Confidential to Editor” section, and submit your "Accept" recommendation.

Reviewer #3: All comments have been addressed

Reviewer #4: (No Response)

2. Is the manuscript technically sound, and do the data support the conclusions?

Reviewer #3: Yes

Reviewer #4: Partly

3. Has the statistical analysis been performed appropriately and rigorously? 

Reviewer #3: Yes

Reviewer #4: (No Response)

4. Have the authors made all data underlying the findings in their manuscript fully available?

Reviewer #3: Yes

Reviewer #4: Yes

5. Is the manuscript presented in an intelligible fashion and written in standard English?

Reviewer #3: Yes

Reviewer #4: No

6. Review Comments to the Author

Reviewer #3: (No Response)

Reviewer #4: I am not convinced my answer still is remain , what is the association between AF and HbA1c ?? Would suggest to have a consultation with an expert in cardiologists or endocrinologist

7. PLOS authors have the option to publish the peer review history of their article (what does this mean?). If published, this will include your full peer review and any attached files.

Reviewer #3: No

Reviewer #4: No

---

## [Editor Report · Acceptance letter]

30 Jan 2020

PONE-D-19-25933R1 

Dose-response analysis between hemoglobin A1c and risk of atrial fibrillation in patients with and without known diabetes 

Dear Dr. xie:

I am pleased to inform you that your manuscript has been deemed suitable for publication in PLOS ONE. Congratulations! Your manuscript is now with our production department. 

With kind regards,

on behalf of

Dr. Ronpichai Chokesuwattanaskul 

Academic Editor

PLOS ONE